# A Deep Learning Architecture Using 3D Vectorcardiogram to Detect R-Peaks in ECG with Enhanced Precision

**DOI:** 10.3390/s23042288

**Published:** 2023-02-18

**Authors:** Maroua Mehri, Guillaume Calmon, Freddy Odille, Julien Oster

**Affiliations:** 1Epsidy, 54000 Nancy, France; 2Ecole Nationale d’Ingénieurs de Sousse, LATIS-Laboratory of Advanced Technology and Intelligent Systems, Université de Sousse, Sousse 4023, Tunisia; 3IADI-Imagerie Adaptative Diagnostique et Interventionnelle, Inserm U1254, Université de Lorraine, 54000 Nancy, France; 4CIC-IT 1433, Inserm, CHRU de Nancy, Université de Lorraine, 54000 Nancy, France

**Keywords:** 12-lead ECG, vectorcardiogram, R-peak detection, segmentation, deep learning, U-Net architecture

## Abstract

Providing reliable detection of QRS complexes is key in automated analyses of electrocardiograms (ECG). Accurate and timely R-peak detections provide a basis for ECG-based diagnoses and to synchronize radiologic, electrophysiologic, or other medical devices. Compared with classical algorithms, deep learning (DL) architectures have demonstrated superior accuracy and high generalization capacity. Furthermore, they can be embedded on edge devices for real-time inference. 3D vectorcardiograms (VCG) provide a unifying framework for detecting R-peaks regardless of the acquisition strategy or number of ECG leads. In this article, a DL architecture was demonstrated to provide enhanced precision when trained and applied on 3D VCG, with no pre-processing nor post-processing steps. Experiments were conducted on four different public databases. Using the proposed approach, high F1-scores of 99.80% and 99.64% were achieved in leave-one-out cross-validation and cross-database validation protocols, respectively. False detections, measured by a precision of 99.88% or more, were significantly reduced compared with recent state-of-the-art methods tested on the same databases, without penalty in the number of missed peaks, measured by a recall of 99.39% or more. This approach can provide new applications for devices where precision, or positive predictive value, is essential, for instance cardiac magnetic resonance imaging.

## 1. Introduction

It is estimated that over 300 million resting ECG tests are performed every year [1], to which exercise ECG tests, and a number of procedures involving ECG synchronization must be added. Such procedures may be imaging the heart or ablating electrical pathways. Over the last few decades, ECG has extensively been studied, focusing on detecting QRS complexes, and classifying features [2]. In an ECG, P-waves indicate atrial depolarization, QRS complexes correspond to ventricular depolarization and T-wave to ventricular repolarization (Figure 1). Automated analyses of ECG signals have been accelerated with the public availability of electronic ECG databases [3].

In ECG, QRS complexes consist of short-time pulses that contain R-peaks with steep slopes and high amplitudes. Algorithms to detect R-peaks and to measure heart rates are ubiquitous in medicine, and automated analyses of ECG are increasingly common. Automatically detecting R-peaks provides essential information to clinicians about the heart activity status. Furthermore, R-peaks play a key role in the subsequent classification of cardiac cycles and identification of abnormalities (e.g., arrhythmia), eventually supporting computer-aided diagnoses [4,5,6].

In this article, our focus is the detection of R-peaks in multi-lead ECG recordings. Arguably, the timings of R-peaks are shifted between leads; however, we aim to define a single R-peak timing representative of the onset of ventricular systole. There has been extensive research focusing on different ECG key features. Besides the detection of R-peaks, the most common approach has been to segment QRS complexes from their onsets to their offsets [6].

In good quality ECG recordings, classical algorithms can detect an R-peak with recall and precision both over 99%. Teaching these algorithms new use cases featuring high levels of noise or artifacts requires adapting their parameters or developing new ones, whereas deep learning (DL) architectures can be tailored through the adaptation of their training databases [7]. Compared with classical algorithms, DL architectures require large amounts of annotated data and high computing resources, which are not available in all circumstances. In the last decade, DL architectures have successfully been explored across many sub-fields following early successes in computer vision and pattern recognition [8,9]. Many studies confirmed that DL architectures outperform classical algorithms for signal analysis, image recognition, and object/pattern detection [10]. Many medical applications have benefited from the development and progress in computing, data availability, and DL to address a large variety of complex challenges [11]. Hence, DL architectures have recently become interesting alternatives to classical ECG signal analysis methods [2,7,12]. Furthermore, DL architectures can automatically identify relationships in data independently from handcrafted features [13]. DL is among the best techniques to detect patterns or to classify objects degraded by different noise levels or types [14]. One of the advantages of DL architectures in the analysis of ECG is that they can easily deal with single or multiple lead ECG recordings [7].

While many R-peak detection techniques have had a strong focus on reducing the number of missed detections (FN), our focus and motivation have been to avoid false detections (FPs). Some diagnostic or therapeutic medical devices use R-peaks as systolic triggers. One instance is cardiac magnetic resonance imaging (CMR): R-peaks trigger the pulse sequence, leading to the acquisition of raw (K-space) data. FPs cause data corruption leading to image artifacts. In noninvasive cardiac radioablations [15], R-peaks’ timings may be used to trigger irradiations of an arrhythmogenic substrate. Reliable R-peak triggers may enhance the precision of radiotherapy beams. Both cases require a low latency, leaving little to no time for a decision-making step. In other imaging, interventional, or robotic fields, high precision cardiac triggers may also be relevant.

## 2. Related Work

In this section, we review the main R-peak detection techniques. Classical algorithms were initially designed with the advent of electronic recordings and computerized analysis. More recently, DL architectures demonstrated groundbreaking performance in detecting R-peaks. This was further accelerated by the public availability of annotated ECG databases, such as MIT-BIH arrhythmia [16,17], and organized challenges, such as CinC [18,19,20,21,22] and CPSC [23,24,25,26]. Finally, hybrid approaches merging DL architectures, signal processing, and structural analysis techniques (e.g., clustering algorithms) have emerged. Methods to detect R-peaks can be categorized into: classical, DL, and hybrid.

### 2.1. Classical Approaches

Classical approaches are based on using various classical signal processing and structural analysis techniques. They usually feature pre-processing and decision-making steps. Classical algorithms are often distinguished according to their pre-processing steps, while the majority of the decision-making steps are either heuristic or based on machine learning (ML), such as support vector machine (SVM) [27] or decision tree algorithms [28], and dependent on pre-processing results.

Pahlm and Sörnmo reviewed pre-1984 methods based on defining rules and signal processing techniques for detecting QRS complexes in one-channel ECG recordings [29]. In 1985, Pan and Tompkins proposed the Pan–Tompkins (PT) algorithm based on using differential thresholds and setting the slope, amplitude, and width of a moving window for locating R-peaks [30]. Noise elimination, signal smoothing, and enhancing width and QRS slope were introduced in a pre-processing step. A decision-making step was used to determine whether or not a detection fits an R -peak. In 1986, Hamilton and Tompkins measured the effects of several parameters (median, iterative, or mean peak estimators) used in the decision-making step and proposed an optimized decision rule to discriminate true R-peak events from false, noise-induced events [31]. In 2022, Khan and Imtiaz introduced Pan-Tompkins++, an improved PT algorithm [32], by adding a filter with a band-pass of 5–18 Hz followed by an N-point moving average filter and different rules to adjust the thresholds based on the signal pattern particularities.

Kohler et al. classified QRS detection methods into several categories based on signal derivatives and digital filters, wavelets, and additional approaches (e.g., hidden Markov models [33]) [34].

Derivative-based algorithms define threshold levels from differentiator filters and difference operation methods [35,36,37,38,39]. For instance, Gutiérrez-Rivas et al. computed a dynamic threshold defined by a finite state machine (FSM) which depends on the sampling frequency of ECG recordings [40]. Typical frequency components of QRS complexes range from ~10 Hz to ~25 Hz. Many sophisticated digital filters have been explored, such as MaMeMi [41] or quadratic [42]. Many linear and non-linear transform operators have been applied in the pre-processing step, such as U3 [43], S [44], Hilbert [45], fast Fourier [46,47], wavelet [48,49,50]. Traditional Fourier or wavelet transform based methods are not suitable for all QRS morphologies: they are sensitive to intra- and inter-subject variations. To address this issue, Zhou et al. proposed a sparse representation-based ECG signal enhancement method that models ECG signals by a combination of inner structures and additive random noise [51] and then used these structures during the training phase to extract original ECG signals and to remove artifacts (e.g., wandering baseline). QRS complexes were identified in enhanced ECG signals as structures having the larger kurtosis values. The Zhou et al. algorithm yielded a high rate of FP [51]. Hossain et al. presented a complete ensemble empirical mode decomposition with adaptive noise (CEEMDAN) for QRS complexes and P-waves detection without using conventional filtering techniques [52]. After carrying a pre-processing step (e.g. baseline drift removal), two different signals are reconstructed by the CEEMDAN method, one for QRS complexes and one for P-waves; the one matching the signal dynamics is selected.

Many algorithms based on the Shannon energy envelope were proposed to detect R-peaks with low delay and high accuracy and speed [53,54], provided that two key thresholds are correctly set. A fast R-peak detection method requiring low resource consumption was presented by Elgendi [55], using two moving averages which are calibrated by means of an optimized knowledge base with two different parameters. For QRS detections, a weighted total variation (WTV) denoising technique was proposed in the pre-processing step [56]. A regularization parameter in WTV minimization and weights is required to adapt locally the amount of applied smoothing and to select QRS complexes over P- and T-waves.

Several classical methods use multiple modalities, combining ECG with other channels, such as blood pressure (BP), electroencephalography (EEG), electrooculography (EOG), electromyography (EMG) and others, to detect R-peaks. For instance, Johnson et al. proposed a method based on defining a signal quality index derived from two different modalities: R-peaks detected in the ECG using energy and R-peaks detected in the arterial BP waveform using length transform [57]. From noisy multi-modal recordings, Gierałtowski et al. used more than two channels (ECG, BP, EEG, EOG, and EMG) for QRS detection [58]. Combining multiple channels requires adjusting rules accordingly. The performance of this approach varies, especially in ECG recordings of short durations.

Liu et al. compared the performances of ten widely used QRS detection algorithms in different cases [59]: PT [30], Hamilton-mean [31], Hamilton-median [31], RS slope [58], sixth power algorithm [37], FSM [40], U3 transform [43], difference operation algorithm [38], window-based peak energy detector [57,60,61], and optimized knowledge base [55]. To guarantee high performance, pre-processing and post-processing steps were usually required, at an increased computing cost. Other reviews and benchmarks focused on pre-processing and decision-making steps [62,63,64]. Van and Podmasteryev reviewed in detail four ML-based algorithms [65], including artificial neural network (ANN) [66], k-nearest neighbor (k-NN) algorithm [67], k-means [68], and SVM [69]. Performances of investigated ML algorithms were shown to depend on the quality of their training datasets, which was an essential drawback.

Few approaches were designed to work with multiple lead ECG recordings. Śmigiel et al. used 6 classical detectors and a k-means clustering method after processing a 12-lead ECG signal [70]. Chen et al. used the PT algorithm to detect R-peaks in each lead of the ECG recordings of the LUDB database [71,72,73].

The main disadvantage of classical algorithms is that they depend on many empirical parameters (e.g., self-adaptive threshold values) and lack robustness, particularly with low-quality, noisy, or pathological signals, changes in sensors or different QRS morphologies. PT for instance, one of the most widely used algorithm to detect R-peaks in commercial cardiac monitoring devices, is sensitive to various sources of noise, including wandering baseline, power-line interferences, muscle artifacts, and electrode contact noise. Moreover, they are often unsuitable for ECG recordings of short durations.

### 2.2. DL-Based Approaches

Several DL architectures have been proposed for different tasks related to computer vision and pattern recognition [8,9], including ECG analysis [2,4]. A DL architecture is an ML algorithm based on a neural network (NN). It works by learning a correlation between input features using a large amount of data. The larger the database, the better the DL architecture can learn by optimizing its weights and perform the targeted task by taking the most suitable decision. Commonly used DL architectures to detect R-peaks are: convolutional neural network (CNN) [74], recurrent neural network (RNN) [75], long short-term memory (LSTM) [76], and gated recurrent unit (GRU) [77].

CNN-based solutions represent the most used techniques for ECG analysis, and particularly for feature extraction tasks [2]. Šarlija et al. reported a 1D CNN-based architecture exceeding 99% in recall and precision on unseen data from the MIT-BIH arrhythmia database [78]. Oudkerk-Pool et al. reported 92.6% recall, 91% precision, and 91.8% F1-score on 100 recordings from the PhysioNet/CinC challenge 2017 using a fully convolutional dilated NN [20,79]. Tison et al. used a modified CNN (U-Net) architecture for automating the classification of six types of ECG segments (P-wave, PR segment, QRS complex, ST segment, T-wave, and TP segment) [80]. Using the MIT-BIH arrhythmia database and a modified U-Net architecture, Oh et al. reported a high recall of 98.76% but a low precision of 29.55% [81]. Jimenez-Perez et al. used a U-Net architecture to analyze dual lead ECG recordings from the QTDB database and achieved a recall of 99.94% [82,83,84].

Some novel DL-based methods combine variant CNN architectures with other DL models. Liu et al. achieved high accuracy by using three variants of bidirectional long short-term memory (BLSTM) architectures: with attention model, combined with U-Net, and combined with U-Net++ [85]. Peimankar and Puthusserypady demonstrated a recall of 97.95% and a precision of 95.68% on 105 recordings from QTDB by combining a CNN architecture with a BLSTM model to detect onset, peak, and offset of different heartbeat waveforms (P-waves, QRS complexes, and T-waves) [86]. Vijayarangan et al. proposed a fused CNN-ResNet, called RPnet, an architecture combining the 1D U-Net model with inception and residual modules, to extract R-peaks from noisy ECG [87]. This approach contributed to alleviate the vanishing gradient issues of DL architectures. RPnet outperformed three classical algorithms (Hamilton and Tompkins [31], Christov [35], and stationary wavelet transform [50]), with an F 1-score of 98.37%, on the second CPSC database [24]. Duraj et al. showed high performance on LUDB by incorporating the residual and squeeze-excitation blocks into the 1D U-Net architecture for extracting segments, such as P-waves, QRS complexes, and T-waves, regardless of the lead [88]. Gabbouj et al. proposed a 1D self-organized operational neural network (ONN), evaluated on second CPSC, and achieved recall of 99.79%, precision of 98.42%, and F1-score of 99.10% [24,89]. However, during the training phase, it featured a high complexity in the number of multiply–accumulate operations and the number of parameters compared with 1D CNN architectures.

Many DL architectures are computationally expensive and hence not well-adapted to embedded devices. Furthermore, CNN and their variants, such as U-Net, are the most common DL architectures for R-peak detection. Yet, many researchers stated that the best accuracy was achieved by combining different DL architectures, at a cost of increasing complexity [90]. Single ECG lead DL architectures already outperform classical algorithms, but few use multiple leads. Increasingly available 12-lead ECG databases may increase the precision and robustness of DL-based approaches.

### 2.3. Hybrid Approaches

To refine the performances of DL-based approaches, signal processing and structural analysis techniques, such as clustering algorithms, ensemble learning methods, and heuristic approaches, can be added into a post-processing step.

Sereda et al. introduced a pre-processing step (wandering baseline removal) [91]. Then, they used a 12 CNN set on LUDB to correct errors occurring in a single CNN. They reported an F 1-score (over 95%) higher than when using a single CNN (94%). In multi-lead ECG recordings, Moskalenko et al. proposed to process the DL output of each lead independently and to average the resulting scores [92]. Their method demonstrated an F 1-score of 99.97%, outperforming other methods, such as the 12-lead set of Sereda et al. [91] or the wavelet-based method of Kalyakulina et al. [12].

To validate detected R-peaks in a single ECG lead from the MIT-BIH arrhythmia and third CPSC databases, Zahid et al. carried out a verification model based on a timing criterion, considering the following hypothesis: if the predicted R-peak locations of two beats fall within 300 ms, one is an F P [25,93].

In multi-lead ECG signals, Han et al. introduced a post-processing step based on an adaptive dynamic threshold strategy and electrophysiology knowledge, after applying a linear ensemble method averaging the outputs of two DL models [94]. They used CNN and LSTM models, both based on the U-Net architecture. A pre-processing step consisted of removing the wandering baseline and high frequency noise based on discrete wavelet transforms and a third order Butterworth band-pass filter. They reported recall and precision, ranging from 99.45% to 100%, on QTDB, LUDB, and CCDD databases [95,96,97]. Their strategy effectively reduced FN and FP; however, it did not correctly detect some arrhythmic QRS complexes, such as left bundle branch blocks.

Cascading several steps of variable complexities may require extensive computational power, and make the application of hybrid approaches difficult in embedded devices. Furthermore, introducing thresholds, heuristics, and rules in post-processing steps may reduce their generalization capacity to other data and reduce their robustness, particularly in the case of pathological QRS morphologies.

## 3. Proposed Approach

The originality of the proposed R-peak detection approach lies in adapting, training, and testing a DL architecture using a 3D VCG instead of a single or multiple lead ECG, without pre-processing or post-processing steps.

### 3.1. VCG Transformation

There is an inherent complexity in ECG signal analysis due to the plethora of configurations of ECG recording electrodes. R-peaks can have positive or negative polarities depending on the lead position, and often, the underlying cardiac pathology. In the 1950s, researchers introduced 3D VCG as a simplified representation of the electrical activity of the heart along three vectors (X, Y, and Z) [98].

Several mathematical transformations allow reconstructing a 3D VCG from a 12-lead ECG [99] based on a matrix product:(1)V=E×M
where *V*, *E*, and *M* denote a 3D VCG vector, a vector representing individual ECG leads, and a transformation matrix, respectively.

Leads III, aVR, aVL, and aVF, obtained by trivial linear combinations of leads I and II, can be eliminated. To convert a 12-lead ECG into a 3D VCG, a transformation was performed according to Equation (Equation 1), where *E* represents the 8 independent ECG leads (I, II, and V1 to V6) and *M* denotes the matrix obtained by Kors regression transformation [100]: (2)$$M=\left(\begin{array}{ccc} \;0.38 & -0.07 & \;0.11\\ -0.07 & \;0.93 & -0.23\\ -0.13 & \;0.06 & -0.43\\ \;0.05 & -0.02 & -0.06\\ -0.01 & -0.05 & -0.14\\ \;0.14 & \;0.06 & -0.20\\ \;0.06 & -0.17 & -0.11\\ \;0.54 & \;0.13 & \;0.31 \end{array}\right)$$

Comparing the Kors quasi-orthogonal transformation with three other VCG transformations (inverse Dower, Kors regression, and Frank’s orthogonal lead system), Kors et al. reported the best results with the Kors regression transformation [100].

Figure 2 illustrates the Kors regression transformation in a recording from the INCART database. In the figures below, all signals were scaled between −1 and 1. All annotated R-peaks were highlighted using two yellow bars delineating a five-sample segment.

### 3.2. DL Architecture

Normalized 3D VCG segments of durations ~4 s (2048 samples at 500 Hz) were fed as input to the DL architecture. A short sample size was decided after initial experiments showing no negative impact on precision. A power of two was required for the encoding/decoding cascade of the DL model. The proposed approach can be performed in any practical durations of ECG recordings, using shorter segments of ~1 s for instance. R-peak times may be shifted between leads, and in the proposed approach, the timings of lead II were used, together with an acceptance window of ±75 ms [59,93,101]. Neither pre-processing nor post-processing steps were used. Hence, the proposed approach can be generalized to low-quality, noisy, or pathological signals acquired from any sensors providing three independent directions of ECG data.

R-peak detection was approached as a segmentation task using the U-Net model. U-Net, used first for 2D medical image segmentation in 2015 by Ronneberger et al., is a DL model following the encoder–decoder architecture and is reputed for its high accuracy and reduced computational complexity [102].

The proposed DL architecture consists of convolutional layers (Conv1D), distributed along two symmetric contracting and expanding paths, which focus on the encoding and decoding processes, respectively. The contracting path focuses on extracting high level abstraction features (context information involving local information) using the convolution and pooling layers to reduce the input dimensionality. The expanding path applies the opposite operations, often called deconvolution (Conv1DTranspose). It is leveraged for precise localization by combining simultaneously global and contextual information captured from the contracting path through skip connections. Indeed, it restores the characteristics of the high level abstraction without information loss using skip connection and up-sampling processes in order to achieve an accurate semantic segmentation task. The bottleneck is the intermediate part between the contracting and expanding paths that contains an encoded information of the input data representing the latent space (relevant information to be able to reconstruct the input data).

Figure 3 illustrates the proposed DL architecture. A detailed description of its layers is provided in Appendix A.

The U-Net model used in this work is composed of multiple hidden layers to guarantee an optimized learning of high-level feature representation of ECG data. It also features a small number of trainable parameters (79,409) compared to other DL models (e.g., transformer [103]). The contracting path down-samples inputs through six convolutional layers with a down-sampling factor of two. The convolutional layers had filters with the same padding. For every 2 consecutive layers, we set the kernel size to 9, 6, and 3 and the number of filters to 16, 32, and 64. Each convolution layer of the last five is followed by a batch normalization layer. A 1D MaxPooling layer was added after each batch normalization layer to down-sample the 1D feature map by a factor of 2. The expanding path decompresses the feature map back to its original size with a reverse configuration of the contracting path. Only the first convolution layer of the expanding path has a dropout layer with a drop rate of 25%. All convolution layers of the proposed DL architecture have LeakyReLU activation functions (with a negative slope coefficient of 25%), except the last one which has a Sigmoid activation to get 1D segmentation maps for R-peaks. We used a filter size of 1 in the convolution operation of the last layer of the expanding path to output a 1D segmentation map. The proposed DL architecture outputs the probability value for every sample point, from which the R-peaks in ECG were centered in a five-sample segment during the annotating process.

To overcome issues related to missed detections of R-peaks located on segments borders, a stride of ~2 s (1024 samples at 500 Hz) was used when extracting 3D VCG segments of ~4 s. Thus, we computed average predictions from overlapping predictions for every sample point of the original signal. Afterward, we filtered predictions by selecting those that were above the probability threshold of 50%. Selected predictions were subsequently adjusted to coincide with a local maximum. Finally, we retained R-peaks if at least five sample points had the same local maximum.

The parameter settings of our DL architecture were first determined based on recently published works [93], and empirically validated. Multiple experiments were carried out, and the best parameters for our setup were retained.

## 4. Experiments

To evaluate the performance of the proposed approach, experiments were conducted on 4 publicly available 12-lead ECG databases, using leave-one-out cross-validation and cross-database validation protocols.

### 4.1. Experimental Corpora

In the context of ECG analyses, not all available databases provide expert-validated annotations of R-peaks locations. Furthermore, public databases are often built using single or dual lead ECG recordings, such as MIT-BIH arrhythmia [16,17], QTDB [83,84], and third CPSC [25] databases.

PTB Diagnostic ECG [104], PTB-XL [105,106], INCART [21,107], first CPSC [23,108], SPH [109], LUDB [72,73], CSE [110], CCDD [95,96,97], and Georgia [111] feature 12-lead ECG recordings. However, the number of publicly available databases having both 12-lead ECG recordings and expert-validated annotations is limited. To the best of our knowledge, CCDD, LUDB, and INCART are the only publicly available databases with the 12 conventional leads (I, II, III, AVR, AVL, AVF, and V1 to V6) and expert-validated R-peak annotations. These 3 databases, more precisely, all recordings from LUDB and INCART, and the first 251 recordings from the 943 publicly available recordings from CCDD were used in our experiments. Another 103 recordings were pseudo-randomly extracted from the remaining CCDD recordings to create an additional database (CCDD-Extra) which was never seen in the training phase of our DL architecture.

1.Chinese Cardiovascular Disease Database (CCDD): is composed of 193,690 12-lead ECG datasets collected and annotated by 2 cardiologists [95,96,97]. Only 943 complete annotated recordings are publicly available. The positions of the QRS complexes, onsets, peaks, offsets for P and T-waves were provided. Among the 943 recordings, the first 251 recordings are usually selected by researchers in their experiments, since the corresponding annotations were made following the same protocol [94]. Based on the set segment size and stride, 753 windows with a total of 3973 R-peaks were obtained and used in our experiments. To appropriately evaluate the proposed approach and conduct an objective comparison with other methods, few adaptations of the annotations of QRS complexes were required. In the first and/or last cycles of the recordings, a few QRS complexes were not annotated. We discarded them, considering only annotated cycles. All samples located inside a QRS interval were annotated, while the proposed approach required only one R-peak location. A k-means algorithm was hence applied to extract one precise R-peak location per QRS interval after setting the number of R-peaks. Figure 4 illustrates a 12-lead ECG recording example of the CCDD database.2.Lobachevsky University Electrocardiography Database (LUDB): is composed of 200 12-lead ECG recordings with a duration of 10 s, and a sampling rate of 500 Hz, recorded using a Schiller Cardiovit AT-101 cardiograph (Schiller AG, Baar, Switzerland) and released on the PhysioNet Website in 2020. It features a large variety of QRS morphologies, from healthy volunteers to patients with different cardiovascular diseases, some of them with pacemakers. All boundaries and peaks of P-waves, QRS complexes and T-waves were manually annotated by cardiologists for each of 12 leads independently. We considered the annotations of lead II in our experiments. Based on the set segment size and stride, 400 windows with a total of 1831 R-peaks were used. In the first and last cycles of the recordings, some R-peaks were not annotated. We discarded them, considering annotated cycles only during the evaluation of the proposed approach. Figure 5 illustrates a 12-lead ECG recording example of LUDB.3.St. Petersburg Institute of Cardiological Technics 12-lead Arrhythmia Database (INCART): is a public 12-lead ECG database sampled at 257 Hz and released on the PhysioNet Website in 2008. It comprises 75 half-hour recordings extracted from 32 Holter recordings which were collected from patients undergoing tests for ischemia, coronary artery disease, conduction abnormalities, and arrhythmia. Based on the set analysis window size and stride, 32,925 windows with a total of 175,907 R-peaks were used in our experiments. Annotations were automatically produced by using an algorithm that detected beat annotations from all 12 leads in the middle of the QRS complexes, and then a few automatic annotations were corrected manually. Hence, there are misaligned annotations in INCART. Figure 6 illustrates a 12-lead ECG recording from INCART.4.CCDD-Extra: is a set of 103 recordings pseudo-randomly selected from the 692 (943−251) remaining recordings of the CCDD database. Annotations of these recordings followed the same protocol as CCDD (i.e. extracting the positions of R-peaks with the k-means algorithm). Based on the set analysis window size and stride, 309 windows with a total of 1616 R-peaks were used. Figure 7 illustrates a 12-lead ECG recording example of the CCDD-Extra database. CCDD-Extra was used to evaluate the performance of our DL architecture in a cross-database validation protocol.

Table 1 summarizes the features of the four databases used in our experiments.

### 4.2. Experimental Protocol

After converting the ECG recordings into 3D VCG, a set of thorough experiments were conducted to evaluate and validate the proposed approach. They followed the experimental protocols described below:1.Leave-one-out cross-validation protocol: was applied to the CCDB, LUDB, and INCART databases independently. One recording was extracted for the evaluation phase, while the remaining recordings of the same database were used for the training phase. For instance, in the case of the INCART database, 74 recordings were used to train our DL architecture, and the 75th remaining one was used for evaluating its performance. Then, an average of the 75 different evaluation experiments was computed.2.Cross-database validation protocol: was carried out by evaluating the proposed approach on a database different from those used in the training phase to demonstrate its generalization capacity [112,113]. In our experiments, all 526 recordings (251+200+75) of CCDB, LUDB, and INCART databases were used to train the DL architecture, while the 103 recordings of CCDD-Extra were used for its evaluation. A total of 34,078 windows and 181,711 R-peaks were used during the training phase.

For both protocols, the same hyperparameters, summarized in Table 2, were selected. For stochastic optimization, an Adam optimizer was used with an initial learning rate of 0.001 and a batch size of 64. Weights were randomly initialized with the Xavier uniform distribution. Training was stopped early only when no more improvement in the cross-entropy loss value was recorded for at least 10 epochs (i.e. patience of 10).

Python version 3.10 was used to implement the proposed approach, and Keras version 2.9 was employed for training the DL architecture. Our experiments were carried out on a physical computer equipped with 2.4 GHz Intel® Core™ i9-12900F having 32 GB of RAM and a single NVIDIA® GeForce RTX™ 3060 with 12 GB of RAM. Google Colaboratory servers were also used (Tesla T4 GPU with 16 GB of RAM). The training phase was processed using CUDA kernels, while the testing phase was implemented with a single CPU. Training on all data from CCDD, LUDB, and INCART took 7274 s on the computer (resp. 4297 s on Google Colaboratory servers). As an example, testing the proposed approach on a ~13 s recording (6484 samples at 500 Hz) of CCDD-Extra took 1.15 s on the computer (0.555 s on Google Colaboratory servers) to detect 13 R-peaks.

Figure 8a–c show the loss curves obtained using the leave-one-out cross-validation protocol. Figure 8d shows the loss curve obtained using the cross-database validation protocol.

All 4 learning curves in Figure 8 demonstrate a good fit as they decrease to a point of stability (500 epochs for both CCDD and LUDB, 78 epochs for INCART, and 479 epochs for the three databases). Moreover, the DL architecture trained on INCART shows superior loss convergence properties (i.e. speed) compared to using CCDD or LUDB. This can be explained by the fact that data in INCART were most representative to suitably train a DL architecture: its statistical characteristics were correctly captured. Indeed, the number of annotated R-peaks in INCART is ~44× (resp. ~96×) larger than in CCDD (resp. LUDB). Furthermore, due to the data heterogeneity yielded by combining the three different databases during training, the minimum loss value (0.00781) is higher in the cross-database validation protocol compared to the loss values obtained in the leave-one-out cross-validation protocol (Figure 8).

To validate the 3D VCG approach, the performances of individual lead-by-lead trainings of the DL architecture were assessed, first using representative ECG leads (I, II, and V1 to V6), and second, using individual 3D VCG vectors (X, Y, and Z). Multi-lead trainings were performed using all 8 ECG leads (I, II, and V1 to V6) as input and the 3 best leads from the first experiment.

### 4.3. Performance Evaluation Metrics

To quantify the performance of the proposed approach and compare it with other methods, three standard performance evaluation metrics, defined below, were adopted [6].
1.Recall (or sensitivity): evaluates the rate of defined R-peak annotations that are correctly predicted.
(3)$$\mathrm{Recall}\left(\%\right)=\frac{\mathrm{TP}}{\mathrm{TP}+\mathrm{FN}}\times 100$$2.Precision (or positive predictive value): evaluates the rate of predicted R-peaks that are correctly matched to defined annotations.
(4)$$\mathrm{Precision}\left(\%\right)=\frac{\mathrm{TP}}{\mathrm{TP}+\mathrm{FP}}\times 100$$3.F1-score: corresponds to a harmonic mean of recall and precision.
(5)$$F1-\mathrm{score}\left(\%\right)=\frac{2\times \mathrm{Precision}\times \mathrm{Recall}}{\mathrm{Precision}+\mathrm{Recall}}\times 100$$
where TP, FN, and FP denote the number of true positives, false negatives, and false positives, respectively. Since the maximum duration of QRS complexes is 150 ms, a tolerance of ±75 ms of the annotated R-peak location was considered when counting TP, FN, and FP [59,93,101].

## 5. Results

The performance of the proposed approach was evaluated on 4 different public 12-lead ECG databases.

### 5.1. Qualitative Results

Figure 9, Figure 10, Figure 11 and Figure 12 illustrate the R-peaks detected using the proposed approach on recordings from CCDD, LUDB, INCART, and CCDD-Extra, respectively. Detected R-peaks are marked with red triangles.

Visual inspection of the qualitative results revealed that the proposed approach for detecting R-peaks is reliable and robust. First and/or last cardiac cycles were sometimes not annotated in some ECG recordings and were discarded from the analysis, as can be seen in Figure 9 and Figure 10. Figure 11 demonstrates that the proposed approach succeeded in detecting the R-peak located between 8 s and 9 s, despite a different QRS morphology compared with its predecessors and successors. In Figure 12, the proposed approach detected all R-peaks from an ECG recording of the CCDD-Extra database, never seen during the training phase. This reproducible result, along with quantitative results described below, confirmed that the proposed approach has the capacity to be generalized.

To illustrate some failed detections, a recording from the INCART database shows noice-induced FP in Figure 13. Another source of FP is pacemaker patients (LUDB). However, these errors were limited as shown in the quantitative analysis. Missed detections (FN) are mostly related to pathological QRS morphologies that are less often represented in the training datasets. They were mostly seen in CCDD and INCART databases.

### 5.2. Quantitative Results

Since a visual assessment of the effectiveness and robustness of a method is inherently subjective, we evaluated the accuracy of the locations of detected R-peaks by computing the three metrics described in Section 4.3. Numbers of TP, missed beats (FN), and false detections (FP) are reproduced when relevant.

#### 5.2.1. Quantitative Results on CCDD, LUDB, INCART, and CCDD-Extra Databases

In the leave-one-out cross-validation protocol, the proposed approach produced results with recall ranging from 99.39% to 99.84% and precision ranging from 99.88% to 100.00% (Table 3). In the cross-database validation protocol, the proposed approach produced results with recall of 99.41% and precision of 99.89% (Table 4).

#### 5.2.2. Training on 3D VCG vs. Single Lead, Single VCG Vector, Three Best Leads, or Eight Leads

The DL architecture was trained on 3D VCG data and could also be trained on a different number of leads, extracted or reconstructed from the 12-lead ECG recordings. Four different experiments were carried out, using the leave-one-out cross-validation protocol and the INCART database. Results are summarized in Table 5.

In the first and second experiments, the DL architecture was trained on one lead/vector at a time. Independent leads (I, II, and V1 to V6) and 3D VCG vectors (X, Y, and Z) were used. The testing of the obtained models, following the leave-one-out cross-validation protocol, consisted in 75×(8+3) trainings and as many experiments.

None of the models trained on independent ECG leads (I, II, and V1 to V6) outperformed recall or precision of the DL architecture trained on combined 3D VCG data. In single lead training, recall and precision were highest with leads II, V2, and V5.

When using individual X, Y, and Z vectors for training, results were good, with recall ranging from 99.39% to 99.82% and a precision of 99.98%. It was interesting to see that models trained on the X, Y, and Z vectors were on par with the most effective individual leads from the experiments above, with X and Y vectors providing slightly better results. The 3D VCG trained model outperformed any of the models trained on individual X, Y, and Z vectors.

A third experiment, called the 03-leads experiment, consisted of training the DL architecture with leads II, V2, and V5 combined. These leads were chosen because they provided the best results in the first experiment. The 03-leads experiment demonstrated a lower number of FP than the 3D VCG trained model (9 vs. 16). However, a larger number of FN were observed (393 vs. 276). Comparing the performances of the 03-leads experiment with those of the 3D VCG trained model, recall was 99.77% vs. 99.84%, and precision 100.00% vs. 99.99%.

A fourth experiment, called the 08-leads experiment, consisted of training the DL architecture with the combination of all 8 independent leads. The 08-leads experiment demonstrated a larger number of FP (25 vs. 16), and the number of FN were 6× higher (1633 vs. 276). Comparing the performances of the 08-leads experiment with those of the 3D VCG trained model, recall was 99.14% vs. 99.84%, and precision was 99.99% in both cases.

#### 5.2.3. Comparison with the PT Algorithm

To compare our results with the PT algorithm, we selected an open-source implementation from [114]. Results from PT were compared with the results of the proposed approach in the leave-one-out cross-validation protocol (Table 6).

Regarding missed detections (FN) and recall, the PT algorithm performance was superior to the proposed approach in several leads (II, V2, V3, V5, and V6) for CCDD, in one lead (V3) for LUDB, and was inferior in all leads for INCART. Regarding false detections (or FP) and precision, the PT algorithm performance was inferior to the proposed approach, regardless of the chosen lead. Average PT values of recall were comparable or inferior to the proposed approach (97.73% to 99.50% vs. 99.39% to 99.84%). Average PT values of precision were always inferior to the proposed approach (92.31% to 99.14% vs. 99.88% to 100.00%).

Results from PT were also compared with the results of the proposed approach in the cross-database validation protocol (Table 7). This is more fitting than the previous comparison as in both cases, the algorithms used to analyze the data were not previously exposed to them. The PT algorithm outperformed the proposed approach in recall in all leads except V6 (99.84% vs. 99.41%). However, its precision was inferior, regardless of the chosen lead (98.42% vs. 99.89%). The two best leads for precision were leads II and V4 (98.96%), exhibiting 16 FP vs. 2 for the proposed approach.

#### 5.2.4. Comparison with Other Methods

In the literature, few R-peak detection methods were evaluated using the same databases. We selected recent methods to compare their performances with ours [71,94,115].

In CCDD, the hybrid method from Han et al. [94] outperformed the proposed approach in recall (99.99% vs. 99.39%), but its precision was inferior (99.45% vs. 100.00%) (Table 8).

In LUDB, the classical algorithm from Chen et al. [71] outperformed the proposed approach in recall (100.00% vs. 99.75%) and was equivalent in precision (99.86% vs. 99.88%) (Table 9).

In INCART, the performance of the classical algorithm from Schmidt et al. [115] was inferior to the proposed approach both in recall (99.43% vs. 99.84%) and in precision (99.91% vs. 99.99%) (Table 10).

## 6. Discussion

This work proposes a DL-based approach to detect R-peaks. The novelty of the proposed approach consists in using reconstructed 3D vectorcardiogram (VCG) leads [100] as input instead of single or multiple ECG lead(s). A 3D VCG is obtained by a linear combination of multiple ECG leads. It reduces the computational complexity (i.e. memory and time requirements) both for training and inference compared to using all available leads. Using a 3D VCG as input during the training phase of a DL architecture has the advantage that both strong temporal correlation features and fine morphological features are extracted. Indeed, a 3D VCG provides a calibrated maximal QRS vector within a specific timing. In contrast, when using multiple ECG leads, each scalar lead has its own sensitivity and timing for the R-peak maximum, which could bias data and subsequently results [116].

The proposed approach has an enhanced precision compared to other methods. The number of false detections (FP) was low without the use of any post-processing step. The number of missed beats (FN) yielded a recall on par with or better than published classical algorithms, DL architectures, or hybrid approaches. Without any post-processing step, the proposed approach outperformed the PT algorithm in precision, dividing the number of FP by a factor eight or more. Reducing FP has an important implication when synchronizing diagnostic or therapeutic devices which rely on a systolic trigger, and assume the heart to be in a given phase to perform certain tasks. In CMR, triggering data acquisition by a false detection could lead to severe imaging artifacts impairing proper image interpretation [117,118].

The proposed approach could have applications in automated measurements of clinical features [119]. For instance, heart rates and heart rate variability measurements were computed based on R-peak detections, using the leave-one-out cross-validation protocol in the INCART database [120]. The obtained values are almost identical to values obtained from ground truth annotations (see Table 11). This is further evidence of the clinical validity of the proposed approach.

The proposed approach was thoroughly evaluated on multiple 12-lead ECG databases in leave-one-out cross-validation and cross-database validation protocols. The cross-database validation protocol, exposing the proposed approach to data never seen in the training phase, demonstrated a tiny drop in performance: recall dropped from 99.66% to 99.41% and precision from 99.96% to 99.89%. This shows the robustness of our approach, acquired from training on a large number of ECG cycles (181,711). However, the databases used for training were unbalanced. The INCART database features a larger number of ECG cycles (175,907) than LUDB (1831) or CCDD (3973); it also has less patients: 75 vs. 200 (LUDB) or 251 (CCDD). More balanced ECG databases may further improve the results and robustness of the trained model. It could also provide more insights to fine-tune the model hyperparameters. The CCDD-Extra database used in the cross-database validation protocol was not completely independent from the CCDD database used for training. It was acquired using the same technique as CCDD, yet consisted of different patients. Testing the proposed approach on ECG databases acquired with different parameters may provide additional insights on its accuracy.

The hybrid method from Han et al. [94] and the classical algorithm from Chen et al. [71] outperformed the proposed approach. They used pre-processing or post-processing steps. Han et al. applied a post-processing step based on electrophysiology knowledge, after combining CNN and LSTM to detect QRS complexes [94]. It reduced the number of missed and false detections at the cost of an increased complexity. Chen et al. applied several pre-processing steps, including an adaptive PT algorithm, to determine the ranges and locations of QRS complexes in each lead and to remove false QRS locations taking other leads into account [71]. The recall of the proposed approach was comparable but slightly inferior to these two methods; however, its precision exceeded theirs.

Qualitatively examining the results, we observed that missed beats (FN) were mostly attributed to extrasystoles. This can be explained by the insufficient statistical representations of such QRS morphologies in the training databases. Zahid et al. identified this behavior and used a data augmentation strategy, generating additional extrasystole segments to address issues related to limited training examples [93]. False detections (FP), although extremely rare, usually happened in situations of noise, artifacts, or pacemaker pulses. They could be dealt with in a post-processing step, which could also reduce the number of FN. However, avoiding any post-processing is an advantage, especially for applications requiring a low latency (e.g., CMR).

Our hypothesis was to use a 3D VCG reconstruction scheme to avoid the arbitrary decision of selecting which lead(s) to use. A 3D VCG can be reconstructed from many ECG acquisition schemes and provides increased robustness to a large variety of changes, such as errors in positioning electrodes, artifacts, acquisition protocols, and devices. Results show that a 3D VCG is an optimal strategy in our application. The experiments resulted in one unexpected outcome that 3D VCG is even better than using all eight independent leads from the full ECG signal. A possible, although partial, explanation is that the noise or artifacts from individual leads are smoothed out in a 3D VCG. Another one is the weighting of individual leads: in a 3D VCG, the precordial leads are captured in the Z vector with a weight of 2/3, whereas in the 08-leads experiment, their weight is 6/8. Scaling down on the number of dimensions is an advantage to reduce training and testing complexity. A 3D VCG seems to provide a good compromise and avoids the arbitrary decision to pick the best lead(s). It can potentially improve the automaticity of detecting R-peaks in a variety of use cases.

Another hypothesis was to define a single QRS label (or timing) representative of the onset of ventricular systole. We used the expert-validated annotations from lead II as the gold standard. This choice may have had a negative impact on the accuracy of the PT algorithm in other ECG leads. It could also have reduced the performance of the DL architecture after training on other ECG leads. Finally, it may explain why 3D VCG vectors X and Y have the highest accuracy. To enhance the performance of our DL model, a refined timing of R-peaks could be modeled on a lead-by-lead basis.

To demonstrate both robustness and generalization capacity of the proposed approach in detecting premature ventricular contraction or supra-ventricular premature beats, we evaluated its performances on the testing set of the MIT-BIH arrhythmia database (DS2) after training it using lead II data from INCART only and compared them with other state-of-the-art R-peak detection methods (Table 12).

The DL architecture used in this work, a modified U-Net, has the potential to run with low latency [93]. It could be implemented on a powerful computer or on a GPU-capable edge device, and run in quasi real-time. Since it does not require CPU intensive pre-processing nor post-processing steps, it could be applicable in a number of use cases. Furthermore, adapting the training database, and thus the trained model, could provide similar accuracy in various use cases. One example of use cases is CMR, where the ECG signal suffers from a magneto-hydrodynamic artifact [121,122]. This artifact could be added to the training datasets. Other use cases featuring high noise could also be addressed by adding similar noise features to the training datasets. A performance evaluation of the impact of adding synthetic noise using a signal to noise ratio metric could be developed to assess the robustness of the proposed approach [123].

## 7. Conclusions and Further Work

In this article, we propose a novel approach for detecting R-peaks based on training a DL architecture with 3D VCG data, without pre-processing nor post-processing steps. R-peak detection was approached as a segmentation task using the U-Net architecture which was fed with normalized 3D VCG segments of durations ~4 s (2048 samples at 500 Hz). The proposed approach can be performed in ECG recordings of any practical durations (e.g., from tens of minutes down to a few seconds) and is robust to low-quality, noisy, or pathological signals acquired from different sensors. It is also robust to various QRS morphologies.

Using the proposed approach, high performances were achieved in leave-one-out cross-validation and cross-database validation protocols. Furthermore, false detections (FP) and missed beats (FN) were significantly reduced compared to recent state-of-the-art methods or the PT algorithm. This approach could be adapted for several use cases and seems relevant to synchronize medical devices with R-peaks with a low latency.

In future work, we intend to embed the proposed approach on different hardware backends (e.g., Raspberry Pi, Jetson Nano, …), and to investigate some optimization tools (e.g., TensorRT, Triton inference server, …) for real-time inference on edge platforms. Furthermore, data augmentation strategies could enhance the performance of the proposed approach in some high noise use cases.

## Figures and Tables

**Figure 1 sensors-23-02288-f001:**
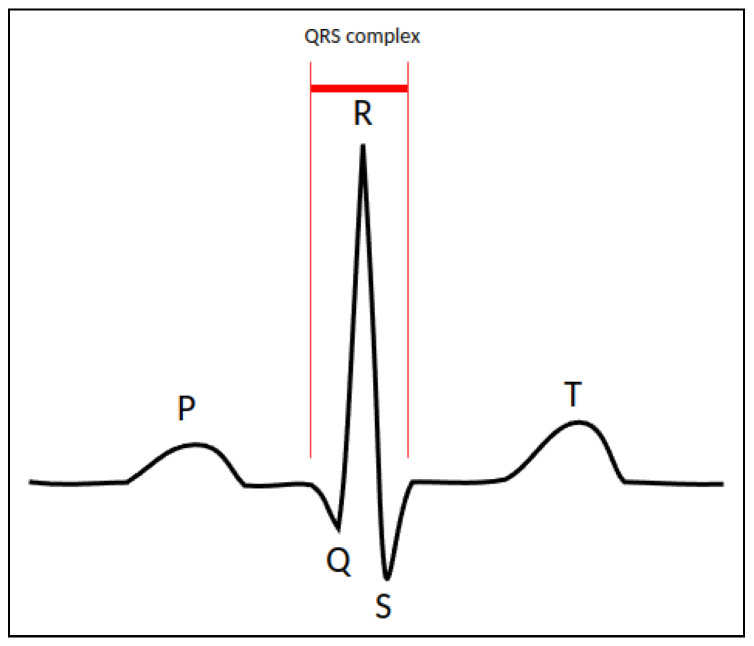
Illustrative schema of the ECG waveform components. P: atrial depolarization; QRS: ventricular depolarization; T: ventricular repolarization.

**Figure 2 sensors-23-02288-f002:**
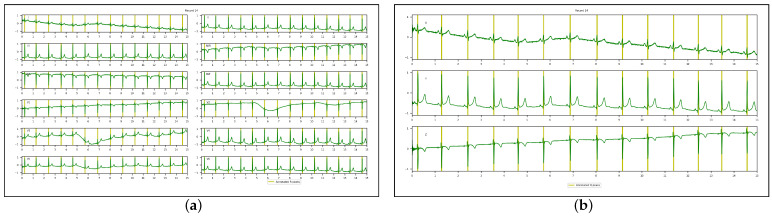
Kors regression transformation of a 12-lead ECG into a 3D VCG in a recording from the INCART database: (**a**) 12-lead ECG; (**b**) 3D VCG.

**Figure 3 sensors-23-02288-f003:**
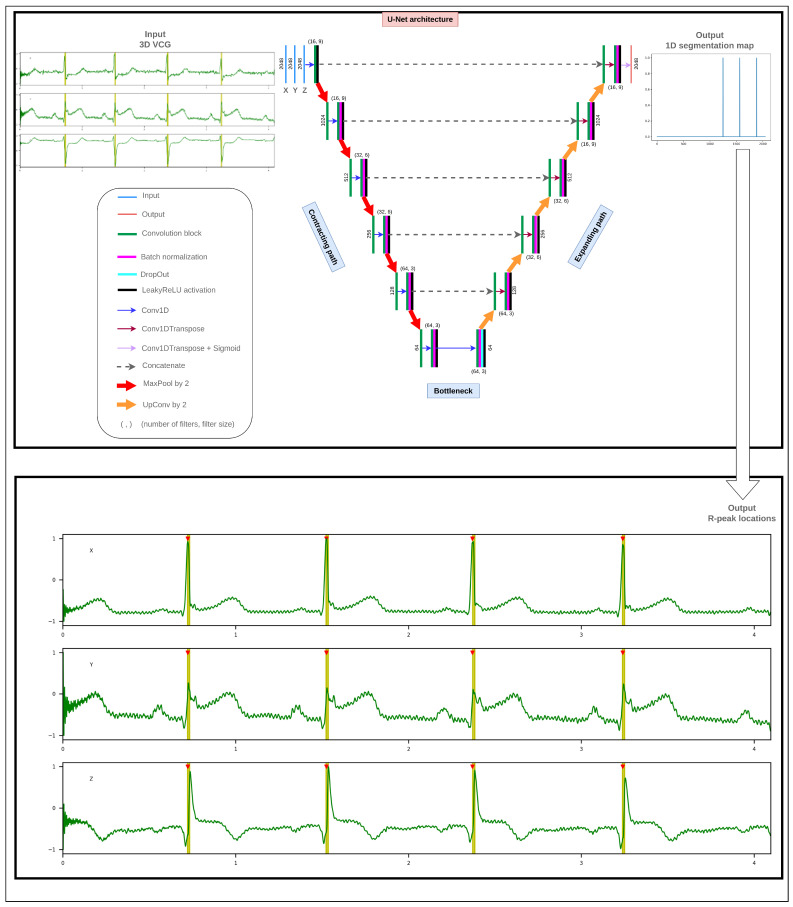
Schema of the proposed DL architecture for R-peak detection.

**Figure 4 sensors-23-02288-f004:**
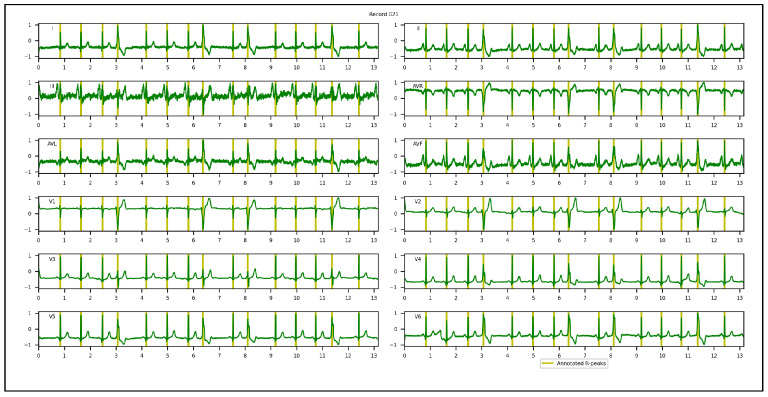
A 12-lead ECG recording example of the CCDD database.

**Figure 5 sensors-23-02288-f005:**
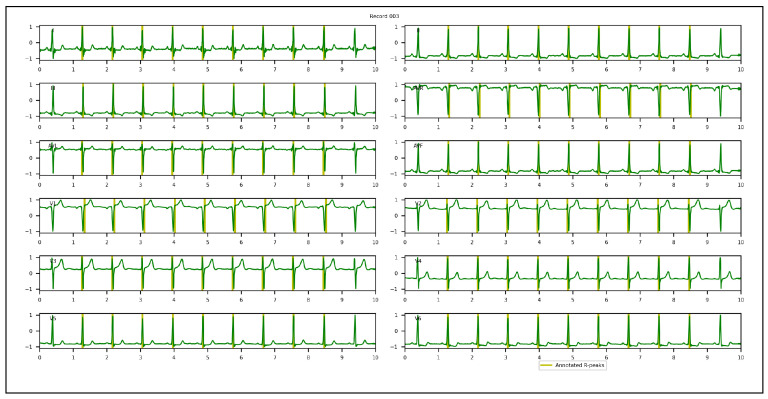
A 12-lead ECG recording example of the LUDB database.

**Figure 6 sensors-23-02288-f006:**
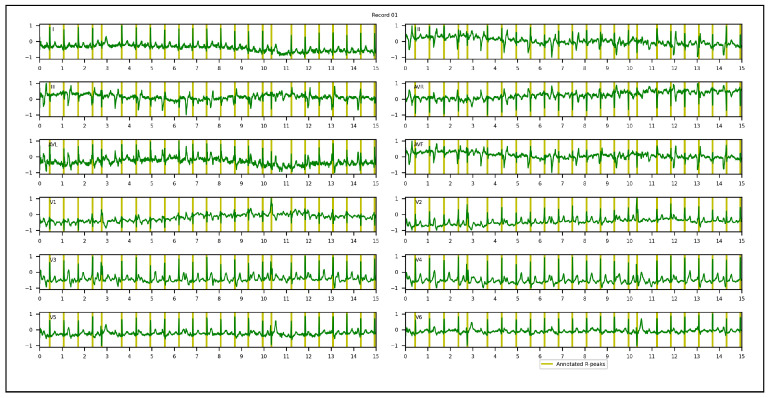
A 12-lead ECG recording from the INCART database.

**Figure 7 sensors-23-02288-f007:**
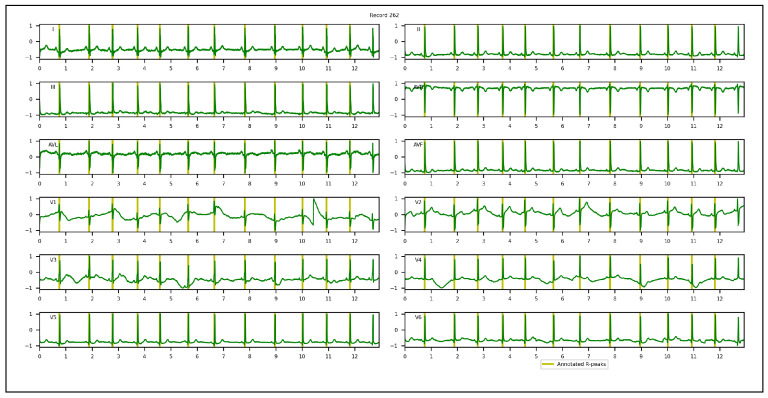
A 12-lead ECG recording example of the CCDD-Extra database.

**Figure 8 sensors-23-02288-f008:**
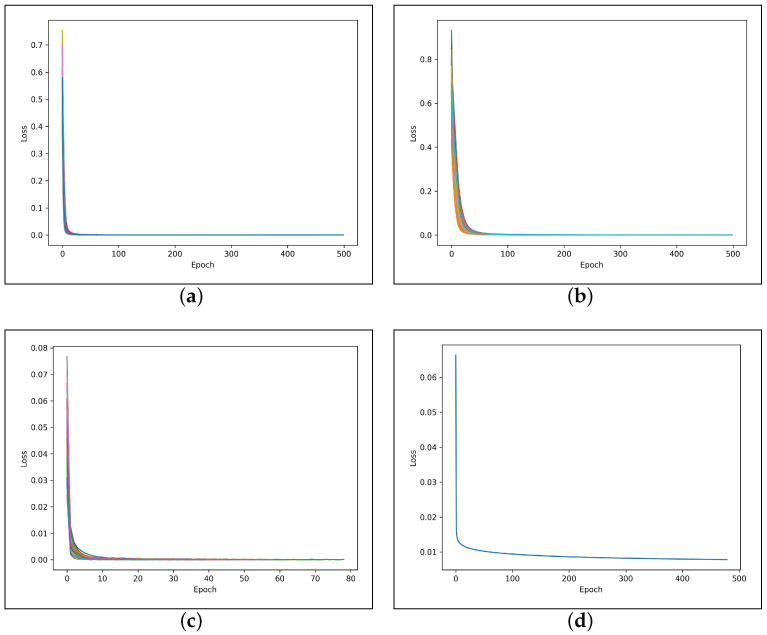
Loss curves obtained using the leave-one-out cross-validation protocol on the training data from CCDD (**a**); LUDB (**b**); and INCART (**c**); loss curve obtained in the cross-database validation protocol using all 526 recordings of the three databases (**d**).

**Figure 9 sensors-23-02288-f009:**
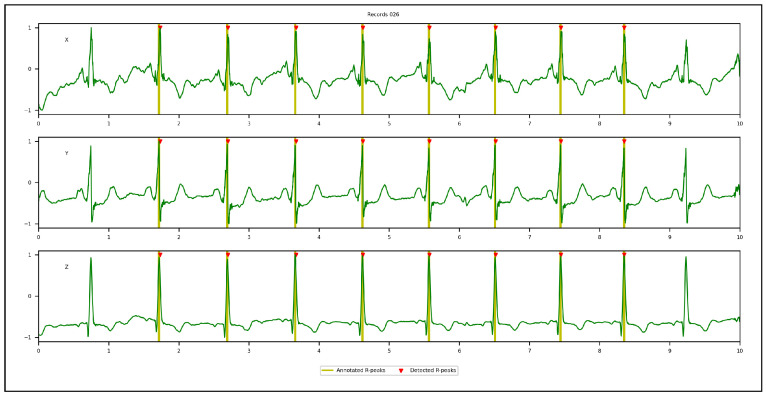
Qualitative results: recording from the CCDD database showing the 3D VCG vectors (X, Y, and Z) and R-peak detected by the proposed approach in the leave-one-out cross-validation protocol. The first and last R-peaks were ignored since there were no annotations.

**Figure 10 sensors-23-02288-f010:**
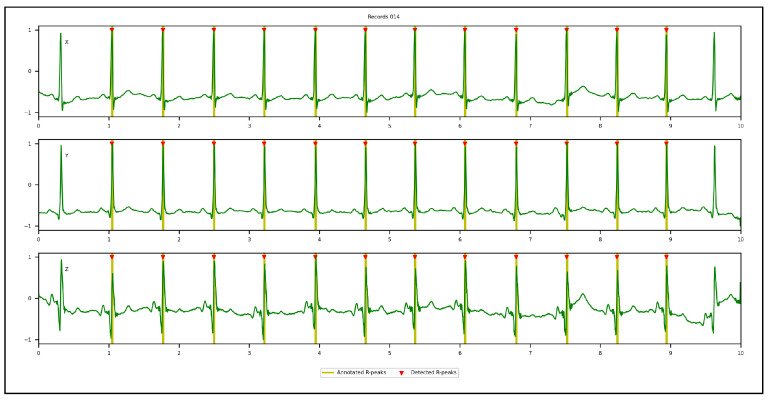
Qualitative results: recording from the LUDB database showing the 3D VCG vectors (X, Y, and Z) and R-peak detected by the proposed approach in the leave-one-out cross-validation protocol. The first and last R-peaks were ignored since there were no annotations.

**Figure 11 sensors-23-02288-f011:**
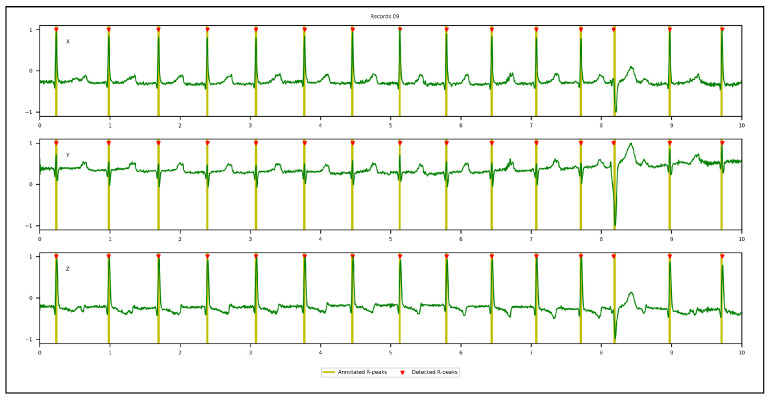
Qualitative results: recording from the INCART database showing the 3D VCG vectors (X, Y, and Z) and R-peak detected by the proposed approach in the leave-one-out cross-validation protocol.

**Figure 12 sensors-23-02288-f012:**
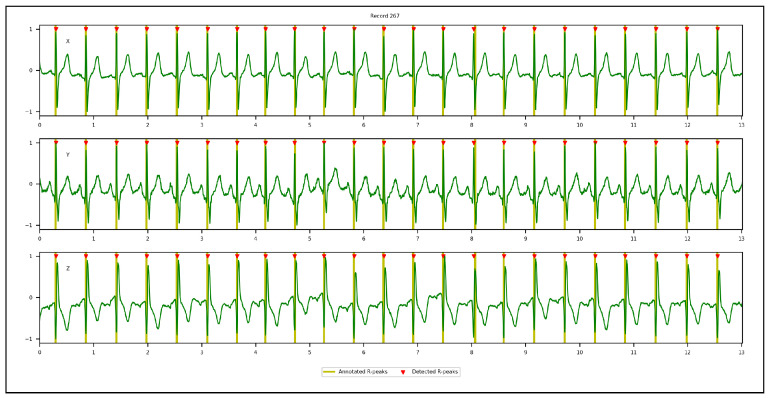
Qualitative results: recording from the CCDD-Extra database showing the 3D VCG vectors (X, Y, and Z) and R-peak detected by the proposed approach in the cross-database validation protocol.

**Figure 13 sensors-23-02288-f013:**
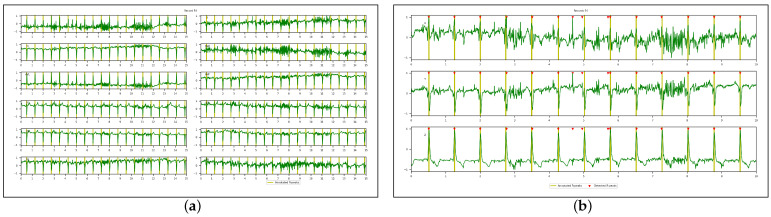
Illustration of a false R-peak detection (FP) when applying the proposed approach on a noisy recording from the INCART database: (**a**) 12-lead ECG; (**b**) 3D VCG.

**Table 1 sensors-23-02288-t001:** Key features of the four databases used in our experiments.

Database	Sampling Frequency (Hz)	Recording Duration	No. of Recordings	No. of Analysis Windows	No. of R-Peaks
**CCDD**	500	~10 s	251	753	3973
**LUDB**	500	10 s	200	400	1831
**INCART**	257	30 min	75	32,925	175,907
**CCDD-Extra**	500	~10 s	103	309	1616
**Total**			629	34,387	183,327

**Table 2 sensors-23-02288-t002:** Selected hyperparameters for training the DL architecture.

Hyperparameter	Value
Optimizer	Adam
Loss function	Binary cross entropy
Weight decay	None
Learning rate	0.001
Number of epochs	500
Patience	10
Batch size	64

**Table 3 sensors-23-02288-t003:** Quantitative results on CCDD, LUDB, and INCART databases using the leave-one-out cross-validation protocol.

Database	TP	FN	FP	Recall	Precision	F1-Score
**CCDD**	3946	27	0	99.39	100.00	99.69
**LUDB**	1827	4	2	99.75	99.88	99.80
**INCART**	175,631	276	16	99.84	99.99	99.91
**Average**				99.66	99.96	99.80

**Table 4 sensors-23-02288-t004:** Quantitative results on the CCDD-Extra database using the cross-database validation protocol.

Database	TP	FN	FP	Recall	Precision	F1-Score
**CCDD-Extra**	1606	10	2	99.41	99.89	99.64

**Table 5 sensors-23-02288-t005:** Quantitative results on the INCART database using the leave-one-out cross-validation protocol and different inputs (single lead, single VCG vector, or multiple leads); 03-leads: II, V2, and V5; 08-leads: I, II, and V1 to V6.

Database	Input		TP	FN	FP	Recall	Precision	F1-Score
**INCART**	**Single lead**	**I**	174,227	1680	606	99.12	99.72	99.40
		**II**	175,516	391	46	99.77	99.97	99.87
		**V1**	171,071	3990	30	97.42	99.98	98.57
		**V2**	174,046	1861	33	98.95	99.98	99.44
		**V3**	175,025	882	4604	99.48	99.09	99.10
		**V4**	174,845	1062	4743	99.40	98.77	98.93
		**V5**	175,241	666	26	99.62	99.99	99.79
		**V6**	174,621	1286	2513	99.34	99.17	99.23
	**Single VCG vector**	**X**	175,602	305	40	99.82	99.98	99.90
		**Y**	175,457	450	36	99.74	99.98	99.86
		**Z**	174,770	1137	38	99.39	99.98	99.68
	**Multiple leads**	**03-leads**	175,514	393	9	99.77	100.00	99.88
		**08-leads**	174,274	1633	25	99.14	99.99	99.55
	**3D VCG**	**X, Y, and Z**	175,631	276	16	99.84	99.99	**99.91**

**Table 6 sensors-23-02288-t006:** Comparison of the performances of the proposed approach with the PT algorithm (single lead) with the leave-one-out cross-validation protocol.

Database	Method	Lead	TP	FN	FP	Recall	Precision	F1-Score
**CCDD**	**PT algorithm**	**I**	3930	43	60	98.99	98.72	98.58
		**II**	3956	17	40	99.61	99.20	99.33
		**V1**	3926	47	52	99.01	98.95	98.72
		**V2**	3967	6	40	99.85	99.09	99.41
		**V3**	3966	7	65	99.86	98.71	99.16
		**V4**	3944	29	52	99.21	98.98	98.94
		**V5**	3954	19	15	99.53	99.61	99.53
		**V6**	3971	2	5	99.95	99.86	99.91
		**Average**				99.50	99.14	99.20
	**Proposed approach**		3946	27	0	99.39	100.00	99.69
**LUDB**	**PT algorithm**	**I**	1812	19	168	99.16	91.70	95.00
		**II**	1817	14	153	99.34	92.36	95.42
		**V1**	1816	15	160	99.09	92.23	95.28
		**V2**	1817	14	148	99.21	92.45	95.42
		**V3**	1829	2	192	99.89	91.54	95.11
		**V4**	1827	4	138	99.78	92.98	96.04
		**V5**	1819	12	132	99.33	92.88	95.84
		**V6**	1806	25	144	98.81	92.37	95.31
		**Average**				99.33	92.31	95.43
	**Proposed approach**		1827	4	2	99.75	99.88	99.80
**INCART**	**PT algorithm**	**I**	167,264	8643	11,581	95.77	93.83	94.65
		**II**	171,920	3987	2660	97.92	98.40	98.02
		**V1**	173,863	2044	1333	98.89	99.34	99.04
		**V2**	174,373	1534	954	99.15	99.46	99.27
		**V3**	171,817	4090	1117	97.85	98.48	97.91
		**V4**	169,830	6077	1210	96.80	98.56	97.26
		**V5**	172,818	3089	776	98.38	99.48	98.82
		**V6**	170,312	5595	844	97.10	98.62	97.58
		**Average**				97.73	98.27	97.82
	**Proposed approach**		175,631	276	16	99.84	99.99	99.91

**Table 7 sensors-23-02288-t007:** Comparison of the performances of the proposed approach with the PT algorithm (single lead) with the cross-database validation protocol (i.e. after training on CCDD, LUDB and INCART databases).

Database	Method	Lead	TP	FN	FP	Recall	Precision	F1-Score
**CCDD-Extra**	**PT algorithm**	**I**	1616	0	68	100.00	97.08	98.30
		**II**	1615	1	16	99.96	98.96	99.44
		**V1**	1615	1	19	99.96	98.77	99.34
		**V2**	1615	1	18	99.96	98,84	99.38
		**V3**	1614	2	41	99.91	98.06	98.83
		**V4**	1615	1	16	99.96	98.96	99.44
		**V5**	1616	0	17	100.00	98.90	99.43
		**V6**	1599	17	20	98.99	97.75	98.35
		**Average**				99.84	98.42	99.06
	**Proposed approach**		1606	10	2	99.41	99.89	99.64

**Table 8 sensors-23-02288-t008:** Comparison of the performances of the proposed approach with a state-of-the-art method in the CCDD database.

Database	Method	Recall	Precision	F1-Score
**CCDD**	**Han et al. [94]**	99.99	99.45	99.71
	**Proposed approach**	99.39	100.00	99.69

**Table 9 sensors-23-02288-t009:** Comparison of the performances of the proposed approach with a state-of-the-art method in the LUDB database.

Database	Method	Recall	Precision	F1-Score
**LUDB**	**Chen et al. [71]**	100.00	99.86	99.92
	**Proposed approach**	99.75	99.88	99.80

**Table 10 sensors-23-02288-t010:** Comparison of the performances of the proposed approach with a state-of-the-art method in the INCART database.

Database	Method	Recall	Precision	F1-Score
**INCART**	**Schmidt et al. [115]**	99.43	99.91	99.66
	**Proposed approach**	99.84	99.99	99.91

**Table 11 sensors-23-02288-t011:** Heart rates and heart rate variability measurements derived from R-peak detections using the leave-one-out cross-validation protocol in INCART: **AVG-HR:** average heart rate; **SD-HR:** standard deviation of heart rates; **RMSSD (ms):** square root of the mean sum of squares of differences between adjacent RR intervals.

	AVG-HR (Beats/min)	SD-HR (Beats/min)	RMSSD (ms)
**Ground truth**	82.18	17.45	200.84
**Proposed approach**	82.07	18.35	201.63

**Table 12 sensors-23-02288-t012:** Performances of some state-of-the-art R-peak detection methods using the MIT-BIH arrhythmia database compared with our DL architecture. Top three lines: classical algorithms; bottom three lines: DL-based architectures; Proposed approach*: our DL architecture trained using only lead II data from INCART.

Database	Method	No. of R-Peaks	Recall	Precision	F1-Score
**MIT-BIH arrhythmia**	**Pan-Tompkins [30]**	109,985	90.95	99.56	95.06
	**Pan-Tompkins++ [32]**	N/R	99.47	99.60	99.54
	**Hamilton and Tompkins [31]**	109,267	99.69	99.77	99.72
	**Habib et al. [112]**	N/R	97.61	91.93	94.68
	**Zahid et al. [93]**	109,475	99.85	99.82	99.83
	**Šarlija et al. [78]**	49,712	99.81	99.93	99.86
	**Proposed approach***	49,712	99.74	99.15	99.44

## Data Availability

The different databases used in this research are publicly available in PhysioNet at https://physionet.org/about/database and http://www.ecgdb.com accessed on 13 December 2022.

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
