# Peer review of "A Deep Learning Architecture Using 3D Vectorcardiogram to Detect R-Peaks in ECG with Enhanced Precision"

_sensors, 2023, doi:10.3390/s23042288_

Round 1

Reviewer 1 Report

I have read the manuscript. It is an interesting work, but it presents some issues that have to be solved. Some comments:

·         Quantitative results should be reported in the Abstract.

·         Introduction should be reviewed: 1)physiological meaning of specific ECG waves should be reported; 2) methods should be reported in the method section and not in the introduction; 3) aim of the paper should be reported at the end of the Introduction section; 4) clinical use and background of the work should be clarified.

·         Transformation of ECG in VCG is a methodological part, thus it cannot be introduced in the “related works” section.

·         The use of vectorcardiography is not clear. How did the authors, after the R-peak identification in VCG leads, obtain the R-peak identification in all 12 leads ECG?

·         Technical choice should be justified. Did the authors apply a parameter tuning?

·         Clinical evaluation should be presented. Specifically, I suggest demonstrating the clinical use of the method by computing heart-rate and heart-rate-variability features (lead-independent features) in order to demonstrate that the clinical use of the method.

·         Did the authors evaluated the dependency of the method to signal-noise-ratio and length of considered signals?

·         Clinical use and clinical interpretability should be stressed in the discussion section.

Author Response

We gratefully acknowledge Reviewer #1 for the quality of proofreading done and their helpful comments. This will be useful in improving our article and guiding us in future work. The manuscript has been rewritten to consider all Reviewer #1’s useful remarks and esteemed suggestions. 
Please see our responses to Reviewer #1’s comments and suggestions in the attached file.

Reviewer 2 Report

1)     The length of Sections 1 and 2 should be reduced. It is too long.

2)     Please explain how the Kors matrix (Equation 2) is obtained.

3)     General comment: In Section 4.1 the authors have mentioned that “All samples located inside the QRS intervals were annotated as R-peak”. If this Is true for CCDD database, then it is odd. Usually the annotated R-peak indicate the precise location (sample number) of an R-peak.

4)     Indeed, the proposed DL-based architecture has been trained on a large number of ECG beats which are collected from a number of different ECG databases. However,

a.      In order to evaluate the performance of an R-peak detection algorithm, MIT-BIH arrhythmia database MUST be used, and the result had to be reported.

b.     A good R-peak detection algorithm should also be able to detect the abnormal ECG-beats such as PVC, APC, fusion beat, supraventricular ECG beats with high accuracy. And, MIT-BIH arrhythmia database contains a lot of these. Therefore, again, testing on MIT-BIH is a must.

c.      The ECG tracings that are shown in the manuscript are comparatively very clean. PTBDB, INCART, LUDB etc. contains less-noisy ECG signals.

d.     After evaluating the performance of the proposed algorithm on MIT-BIH it should be compared to a few (at least 5 or 6) state-of-the-art algorithms which are published in recent years.

Author Response

We gratefully acknowledge Reviewer #2 for the quality of proofreading done and their helpful comments. This will be useful in improving our article and guiding us in future work. The manuscript has been rewritten to consider all Reviewer #2’s useful remarks and esteemed suggestions. 
Please see our responses to Reviewer #2’s comments and suggestions in the attached file.

Round 2

Reviewer 1 Report

I read again the paper. The authors did not provide a point-to-point answer file. Following some comments:

·       Sentences like “The background on which this work was carried out is the following.” should be avoided.

·       Clinical use and Novelty are two different contents. Clinical use should be introduced in order to clarify the problem that authors aimed to solve (and thus the aim), while the novelty should be discussed in the Discussion section. I suggest revising Introduction and Discussion sections accordingly.

·       The description of the content of the sections should be removed.

·       Technical choice should be justified. Why did the authors select these specific parameters? I suggest, at least, to add citations that proved the use of these parameters.

·       Considering the comparison with other methods, the proposed approach is not the best in terms of evaluation metrics. I suggest discussing these results.

·       I was not able to find the answers to the following comments:

o   Clinical evaluation should be presented. Specifically, I suggest demonstrating the clinical use of the method by computing heart-rate and heart-rate-variability features (lead-independent features) in order to demonstrate that the clinical use of the method.

o   Did the authors evaluated the dependency of the method to signal-noise-ratio and length of considered signals?

o   Clinical use and clinical interpretability should be stressed in the discussion section.

Author Response

Please see our responses to Reviewer #1’s comments and suggestions in the attached file.

We gratefully acknowledge Reviewer #1 for the quality of proofreading done and their helpful comments. This will be useful in improving our article and guiding us in future work. The manuscript has been rewritten to consider all Reviewer #1’s useful remarks and esteemed suggestions.

Round #2

Our responses to Reviewer #1’s comments and suggestions of Round #2 are as follows:

--------------------------------------------------------------------------------------------------------------------

Point 1:

I read again the paper. The authors did not provide a point-to-point answer file.

Response 1:

A point-to-point answer to Reviewer #1’s comments and suggestions of Round #1 was presented in a file attachment when submitting the revised manuscript. We are sorry that this document was lost in the review process. Our responses to comments from Round #1 are reproduced below, following our responses to comments of Round #2.

--------------------------------------------------------------------------------------------------------------------

Point 2:

Sentences like “The background on which this work was carried out is the following.” should be avoided.

Response 2:

As suggested by Reviewer #1, the 1st sentence of the 5th paragraph in Section 1. Introduction was removed (Page 2).

--------------------------------------------------------------------------------------------------------------------

Point 3:

Clinical use and Novelty are two different contents. Clinical use should be introduced in order to clarify the problem that authors aimed to solve (and thus the aim), while the novelty should be discussed in the Discussion section. I suggest revising Introduction and Discussion sections accordingly.

Response 3:

We agree with Reviewer #1 that clinical use should be clearly presented in Section 1. Introduction. Our work aimed at defining R-peaks with high precision. It was not intended to provide a clinical application by itself, but to serve as a 1st step for synchronizing other medical devices or be used in subsequent ECG analyses (e.g. beat classification, heart rate estimation). We added the following sentences in the 5th paragraph in Section 1. Introduction:

“While many R-peak detection techniques have had a strong focus on reducing the number of missed detections (FN), our focus and motivation have been to avoid false detections (FP). Some diagnosis or therapeutic medical devices use R-peak as systolic triggers. One instance is cardiac magnetic resonance imaging (CMR): R-peaks trigger the pulse sequence, leading to the acquisition of raw (K-space) data. FP cause data corruption leading to image artifacts. In noninvasive cardiac radioablations [15], R-peaks timings may be used to trigger irradiations of an arrhythmogenic substrate. Reliable R-peak triggers may enhance the precision of radiotherapy beams. Both cases require a low latency, leaving little to no time for a decision-making step. In other imaging, interventional or robotic fields, high precision cardiac triggers may also be relevant.” (Page 2).

As suggested by Reviewer #1, we reorganized both Section 1. Introduction and Section 6. Discussion. Furthermore, we move the description of the novelty of our work in the 1st paragraph of Section 6. Discussion (Page 20).

--------------------------------------------------------------------------------------------------------------------

Point 4:

The description of the content of the sections should be removed.

Response 4:

As suggested by Reviewer #1, the descriptions of the content of the sections were removed.

--------------------------------------------------------------------------------------------------------------------

Point 5:

Technical choice should be justified. Why did the authors select these specific parameters? I suggest, at least, to add citations that proved the use of these parameters.

Response 5:

We agree with Reviewer #1 that we should justify the technical choices and detail how we tuned parameters by adding citations that justified the use of these parameters. We clarified this point in Section 3.2. DL architecture, adding the following sentences:

  • “Normalized 3D VCG segments of durations ~4 s (2,048 samples at 500 Hz) were fed as input to the DL architecture. A short sample size was decided after initial experiments showing no negative impact on precision. A power of 2 was required for the encoding/decoding cascade of the DL model. The proposed approach can be performed in any practical durations of ECG recordings, using shorter segments of ~1 s for instance. R-peak times may be shifted between leads, and in the proposed approach the timings of lead II were used, together with an acceptance window of ±75 ms [61, 95, 102].” (Page 7).
  • “The parameter settings of our DL architecture were first determined based on recently published works [101], and empirically validated. Multiple experiments were carried out, and the best parameters for our setup retained.” (Page 9).

--------------------------------------------------------------------------------------------------------------------

Point 6:

Considering the comparison with other methods, the proposed approach is not the best in terms of evaluation metrics. I suggest discussing these results.

Response 6:

We agree with Reviewer #1 that we should better analyze and discuss the achieved results when comparing them with those of the state-of-the-art. We added the following sentences in Section 6.  Discussion:

“The hybrid method from Han et al. [96] and the classical algorithm from Chen et al. [73] outperformed the proposed approach. They used pre-processing or post-processing steps. Han et al. applied a post-processing step based on electrophysiology knowledge, after combining CNN and LSTM to detect QRS complexes [96]. It reduced the number of missed and false detections at the cost of an increased complexity. Chen et al. applied several pre-processing steps, including an adaptive PT algorithm, to determine the ranges and locations of QRS complexes in each lead, and to remove false QRS locations taking other leads into account [73]. The recall of the proposed approach was comparable but slightly inferior to these two methods, however its precision exceeded theirs.” (Page 21).

--------------------------------------------------------------------------------------------------------------------

Point 7:

Clinical evaluation should be presented. Specifically, I suggest demonstrating the clinical use of the method by computing heart-rate and heart-rate-variability features (lead-independent features) in order to demonstrate that the clinical use of the method.

Response 7:

We agree with Reviewer #1 that the clinical use of our work should be demonstrated. Our work aimed at defining R-peaks with high precision. It was not intended to provide a clinical application by itself, but to serve as a 1st step for synchronizing other medical devices or be used in subsequent ECG analyses (e.g. beat classification, heart rate estimation). We added the following sentences in Section 6. Discussion:

“The proposed approach has an enhanced precision compared to other methods. The number of false detections (FP) was low without the use of any post-processing step. The number of missed beats (FN) yielded a recall on par with or better than published classical algorithms, DL architectures, or hybrid approaches. Without any post-processing step, the proposed approach outperformed the PT algorithm in precision, dividing the number of FP by a factor 8 or more. Reducing FP has an important implication when synchronizing diagnostic or therapeutic devices which rely on a systolic trigger, and assume a heart condition, to perform certain tasks. In CMR, triggering data acquisition by a false detection could lead to severe imaging artifacts impairing proper image interpretation [118, 119].” (Pages 20-21).

--------------------------------------------------------------------------------------------------------------------

Point 8:

Did the authors evaluate the dependency of the method to signal-noise-ratio and length of considered signals?

Response 8:

The concern of Reviewer #1 about the dependency of our results with signals of various lengths and signal-to-noise ratios is understandable.

1) The proposed approach was applied to signals of variable lengths: CCDD ~10 s, LUDB 10 s, INCART 30 min, and MIT-BIH 30 min. As long as the durations of the signals are longer than the length of the extracted segments (~4 s), training and testing phases could operate.

2) The databases on which the proposed approach was evaluated did not display the full noise variability encountered in clinical practice. In Section 6. Discussion (pages 21-22), we added results obtained on the MIT-BIH database; they demonstrate the reliability of the proposed approach in signals featuring different levels of noise. In Figure 13 (Page 16), we illustrated a challenging example of an ECG recording from the INCART database; it shows R-peak detected using the proposed approach, including a couple of false positives, alongside the noise and artifacts demonstrated by the signals of the 12 leads.

Furthermore, we added the following sentences in Section 6. Discussion:

“Other use cases featuring high noise could also be addressed by adding similar noise features to the training datasets. A performance evaluation of the impact of adding synthetic noise using a signal to noise ratio metric could be developed to assess the robustness of the proposed approach [122].” (Pages 22).

--------------------------------------------------------------------------------------------------------------------

Point 9:

Clinical use and clinical interpretability should be stressed in the discussion section.

Response 9:

We agree with Reviewer #1 that more emphasis on clinical use and interpretability could be desired. We added the following sentences in Section 6. Discussion:

 “The proposed approach has an enhanced precision compared to other methods. The number of false detections (FP) was low without the use of any post-processing step. The number of missed beats (FN) yielded a recall on par with or better than published classical algorithms, DL architectures, or hybrid approaches. Without any post-processing step, the proposed approach outperformed the PT algorithm in precision, dividing the number of FP by a factor 8 or more. Reducing FP has an important implication when synchronizing diagnostic or therapeutic devices which rely on a systolic trigger, and assume a heart condition, to perform certain tasks. In CMR, triggering data acquisition by a false detection could lead to severe imaging artifacts impairing proper image interpretation [118, 119].” (Pages 20-21).

--------------------------------------------------------------------------------------------------------------------

We thank Reviewer #1 once again for valuable comments and suggestions. We hope our revision meets their approval.

Round #1

Our responses to Reviewer #1’s comments and suggestions of Round #1 are as follows:

--------------------------------------------------------------------------------------------------------------------

Point 1:

Quantitative results should be reported in the Abstract.

Response 1:

We added F1-score, recall and precision values in the abstract (Page 1).

--------------------------------------------------------------------------------------------------------------------

Point 2:

Introduction should be reviewed: 1) physiological meaning of specific ECG waves should be reported; 2) methods should be reported in the method section and not in the introduction; 3) aim of the paper should be reported at the end of the Introduction section; 4) clinical use and background of the work should be clarified.

Response 2:

1) We reworded Section 1. Introduction, adding the following sentence:

“In an ECG, P-waves indicate atrial depolarization, QRS complexes correspond to ventricular depolarization and T-wave to ventricular repolarization (figure 1).” (Page 1).

2) we summarized Section 1. Introduction, in particular the 4th paragraph (Page 2). We also removed redundant text, to avoid repetitions, and moved methodology items to Section 3.1. VCG transformation (Pages 6-7).

3) & 4) We clarified the aim of the paper and clinical background in Section 1. Introduction, in particular the 6th paragraph as follows:

“The aim of this paper is to propose a novel DL architecture for precise R-peak detection which can serve in applications such as synchronizing diagnostic or therapeutic devices or support other automated analyses such as beat classification.” (Page 2).

We clarified the clinical background introducing it with: “The background on which this work was carried out is the following.” (Page 2).

--------------------------------------------------------------------------------------------------------------------

Point 3:

Transformation of ECG in VCG is a methodological part, thus it cannot be introduced in the “related works” section.

Response 3:

We summarized the 3D VCG transformation and moved it. to Section 3.1. VCG transformation (Pages 6-7).

--------------------------------------------------------------------------------------------------------------------

Point 4:

The use of vectorcardiography is not clear. How did the authors, after the R-peak identification in VCG leads, obtain the R-peak identification in all 12 leads ECG?

Response 4:

We clarified the point on R-peak identification in Section 3.1. VCG transformation (Pages 6-7) with the addition of the following sentence:

“R-peak times may be shifted between leads, and in the proposed approach the timings of lead II were used, together with an acceptance window of +/- 75 ms.”

--------------------------------------------------------------------------------------------------------------------

Point 5:

Technical choice should be justified. Did the authors apply a parameter tuning?

Response 5:

We agree with Reviewer #1 that we should justify the technical choices and detail how we tuned parameters. We clarified this point in Section 3.2. DL architecture, adding the following sentences:

  • “Normalized 3D VCG segments of durations ~4 s (2,048 samples at 500 Hz) were fed as input to the DL architecture. A short sample size was decided after initial experiments showing no negative impact on precision. A power of 2 was required for the encoding/decoding cascade of the DL model. R-peak times may be shifted between leads, and in the proposed approach the timings of lead II were used, together with an acceptance window of ±75 ms.” (Page 7).
  • “The parameter settings of our DL architecture were first determined based on recently published works [101], and empirically validated. Multiple experiments were carried out, and the best parameters for our setup retained.” (Page 9).

--------------------------------------------------------------------------------------------------------------------

Point 6:

Clinical evaluation should be presented. Specifically, I suggest demonstrating the clinical use of the method by computing heart-rate and heart-rate-variability features (lead-independent features) in order to demonstrate the clinical use of the method.

Response 6:

We agree with Reviewer #1 that clinical evaluation should be clearly presented. Our work aimed at defining R-peaks with high precision. It was not intended to provide a clinical application by itself, but to serve as a 1st step for synchronizing other medical devices or be used in subsequent ECG analyses. We added the following sentences in Section 6. Discussion:

“Reducing FP has an important implication when synchronizing diagnostic or therapeutic devices which rely on a systolic trigger, and assume a heart condition, to perform certain tasks. In CMR, triggering data acquisition by a false detection could lead to severe imaging artifacts impairing proper image interpretation [123].” (Page 20).

--------------------------------------------------------------------------------------------------------------------

Point 7:

Did the authors evaluate the dependency of the method to signal-noise-ratio and length of considered signals?

Response 7:

The concern of Reviewer #1 about the dependency of our results with signals of various lengths and signal-to-noise ratios is understandable. The databases on which the proposed approach was evaluated did not fully display the variability which can be encountered in clinical practice. In Section 6. Discussion (pages 21-22), we added results obtained on the MIT-BIH database; They demonstrate the reliability of the proposed approach in signals of varying durations. In Figure 13 (Page 16), we illustrated a challenging example of an ECG recording from the INCART database; it shows R-peak detected using the proposed approach, including a couple of false positives, alongside the noise and artifacts demonstrated by the signals of the 12 leads.

--------------------------------------------------------------------------------------------------------------------

Point 8:

Clinical use and clinical interpretability should be stressed in the discussion section.

Response 8:

We agree with Reviewer #1 that more emphasis on clinical use and interpretability could be desired. We added the following sentence in Section 6. Discussion:

“Reducing FP has an important implication when synchronizing diagnostic or therapeutic devices which rely on a systolic trigger, and assume a heart condition, to perform certain tasks. In CMR, triggering data acquisition by a false detection could lead to severe imaging artifacts impairing proper image interpretation [123].” (Page 20).

--------------------------------------------------------------------------------------------------------------------

We thank Reviewer #1 once again for valuable comments and suggestions. We hope our revision meets their approval.

Reviewer 2 Report

No further comment.

Author Response

We appreciate the positive feedback from Reviewer #2. We would like to thank Reviewer #2 for recommending our article to be accepted.

Round 3

Reviewer 1 Report

I read again the work. Authors solved most of my comments. About the clinical use, I again strongly suggest to compute clinical features that may demonstrate the applicability of the methods in a real clinical scenario.

Author Response

Please see our responses to Reviewer #1’s comments and suggestions in the attached file.

We gratefully acknowledge Reviewer #1 for the quality of proofreading done and their helpful comments. This will be useful in improving our article and guiding us in future work. The manuscript has been rewritten to consider all Reviewer #1’s useful remarks and esteemed suggestions.

Round #3

Our responses to Reviewer #1’s comments and suggestions of Round #3 are as follows:

--------------------------------------------------------------------------------------------------------------------

Point 1:

I read again the work. Authors solved most of my comments. About the clinical use, I again strongly suggest to compute clinical features that may demonstrate the applicability of the methods in a real clinical scenario.

Response 1:

We thank Reviewer #1 for this insight, and as suggested added heart rate and heart rate variability in Table 11. These clinical features were commented in Section 6. Discussion (Page 20).

--------------------------------------------------------------------------------------------------------------------

We thank Reviewer #1 once again for valuable comments and suggestions. We hope our revision meets their approval.